



# The influence of crustal strength on rift geometry and development – Insights from 3D numerical modelling

Thomas B. Phillips[1], John B. Naliboff[2], Ken J. W. McCaffrey[1], Sophie Pan[3], Jeroen van Hunen[1], Malte Froemchen[1]

[1] Department of Earth Science, Durham University, Science Labs, Durham, DH13LE, UK

[2] Department of Earth and Environmental Science, New Mexico Institute of Mining and Technology, Socorro, New Mexico, USA

[3] Basins Research Group (BRG), Imperial College, London, SW72BP, UK

*Correspondence to*: Thomas B. Phillips (tbphil13@gmail.com)

**Abstract.** The lateral distribution of strength within the crust is non-uniform, dictated by crustal lithology and the presence and distribution of heterogeneities within it. During continental extension, areas of crust with distinct lithological and rheological properties manifest strain differently, influencing the structural style, geometry and evolution of the developing rift system. Here, we use 3D thermo-mechanical models of continental extension to explore how pre-rift upper crustal strength variations influence rift physiography. We model a 500x500x100 km volume containing 125 km wide domains of mechanically 'Strong' and 'Weak' upper crust along with two reference domains, based upon geological observations of the Great South Basin, New Zealand, where extension occurs perpendicular to distinct geological terranes and parallel to terrane boundaries. Crustal strength is represented by varying the initial strength of 5 km³ blocks. Extension is oriented parallel to the domain boundaries such that each domain is subject to the same 5 mm/yr extension rate. Our modelling results show that strain initially localises in the Weak domain, with faults initially following the distribution of Initial Plastic Strain before reorganising to produce a well-established network, all occurring in the initial 100ky timestep. In contrast, little to no localisation occurs in the Strong domain, which is characterised by uniform strain. We find that although faults in the Weak domain are initially inhibited at the terrane boundaries, they eventually propagate through and 'seed' faults in the relatively stronger adjacent domains. We show characteristic structural styles associated with 'strong' and 'weak' crust and relate our observations to rift systems developed across laterally heterogeneous crust worldwide, such as the Great South Basin, NZ, and the Tanganyika rift, East Africa.





## 1 Introduction

Continental lithosphere is highly heterogeneous, with distinct areas of relative strength and weakness ubiquitous across
multiple scales of observation (e.g. Thomas, 2006; Kirkpatrick et al., 2013). This non-uniform distribution of strength and
heterogeneity within the crust influences strain localisation, exerting a great influence over the geometry and development of
rift systems developed during continental extension (e.g. Holdsworth et al., 2001; Kirkpatrick et al., 2013; Brune et al., 2017;
Phillips et al., 2019;  Howell et al., 2019; Wright et al., 2020; Gouiza and Naliboff, 2021)

Structures deep within the lithosphere and in the mantle have previously been shown to focus deformation during tectonic
events, for example, controlling the locations of orogenic fronts (Heron et al., 2019). At the large scale, areas of relatively
undeformed cratonic lithosphere are surrounded by relatively highly deformed mobile orogenic belts, which often control the
siting of rift systems (Daly et al., 1989, Schiffer et al., 2020). Similarly, lithospheric thickness, and the vertical stratification
of strength within it, may influence the development of rift systems and continental margins (Duretz et al., 2016; Brune et al.,
2017; Schiffer et al., 2020; Gouiza and Naliboff, 2021; Beniest et al, 2018). However, whilst lithospheric-scale features may
influence first-order rift geometry and evolution, we here focus on how the distribution of strength and heterogeneity within
the upper crust influences rift geometry and evolution, particularly at the basin-scale and during the early stages of rifting.

The upper crust comprises a mosaic of geological bodies and units, each with unique lithologies and tectonic histories.
Typically strong crustal volumes may include rheologically strong cratons or relatively homogeneous granitic batholiths (e.g.
Thomas, 2019; Howell et al., 2020), whilst weak areas may include rheologically weaker sedimentary sequences. Large
strength contrasts exist between these different lithologies, often across short distances as distinct areas of crust are created,
deformed and juxtaposed against one another throughout multiple tectonic events (Thomas, 2006). In addition to their
rheology, heterogeneities associated with prior deformation, such as shear zones within orogenic belts (Daly et al., 1989) or
pre-existing faults within older rift systems (Cowie et al., 2005; Henza et al., 2011; Naliboff and Buiter, 2015) may also
influence bulk crustal strength by acting as focal points for deformation (e.g. Sutton and Watson, 1986; Holdsworth et al.,
2001). Such structures have been shown to reactivate during extension or segment rift systems depending on their orientation
with respect to the regional stress field (Doré et al., 1997; Mortimer et al., 2002; Fossen et al., 2017; Phillips et al., 2019;
Vasconcelos et al., 2019). Whilst rifting can occur in strong cratonic lithosphere (e.g. Larsen et al., 2008; Tiberi et al., 2019),
previous studies have demonstrated that, across multiple scales, strain preferentially localises into relatively weaker areas with
stronger bodies proving resistant to extension (e.g. Beniest et al., 2017; Lang et al., 2020; Wright et al., 2020; Samsu et al.,
2021). However, less is known about how the characteristic geometry and development of fault networks and rift systems
varies across these relatively 'strong' and 'weak' areas during extension.

Our study is primarily influenced by geological observations from the Great South Basin, offshore New Zealand, where rifting
occurred roughly perpendicular to the boundaries between multiple elongate basement terranes of varying lithology (Phillips
and McCaffrey, 2020; Sahoo et al., 2020; Tulloch et al., 2019; Barrier et al., 2020), including the dominantly granitic Median
Batholith and the dominantly sedimentary Murihiku Terrane, a relict forearc basin (Figure 1a) (Campbell et al., 2003; Campbell





et al., 2019). Due to this geometry, where terrane boundaries are oriented parallel to regional extension, each terrane is subject to the same stress, offering insights into how strain is accommodated across areas of differing strength, and the effect of this on rift geometry and development. As well as the bulk strength of the various terranes, the boundaries between them also form prominent upper crustal structures that may be exploited during later tectonic events (Figure 1c) (Mortimer et al, 2002; Tarling

et al., 2019; Phillips and McCaffrey, 2019; Phillips and Magee, 2020). The structural style and evolution of rift systems reflects the geologically and rheologically complex crustal substrate beneath them, yet how strain is manifest across and within these areas of differing strength and lithology remains relatively unknown.

In this study, we use 3D thermo-mechanical simulations of continental rifting to investigate how rift physiography varies across crustal units of varying initial strength and their respective boundaries. We extend a 500x500x100 km region consisting

of four 125-km wide domains, each assigned different crustal strengths and oriented parallel to the extension direction (Figure 1c). The relative strengths of each domain is represented by randomly varying the initial brittle strength (parameterized through plastic strain softening) between 5 $km^3$ 'Unit Blocks', with weaker domains containing weaker Unit Blocks and a greater contrast between blocks. We explore a range of different parameters representing the strength of our various domains, varying the degree of strain weakening and the amount of initial plastic strain within the models. Our modelling results highlight how

crustal strength and heterogeneities related to prior deformation control strain localisation and rift physiography. We document characteristic structural styles associated with strong and weak crust, examine how faults behave at the boundaries between different domains, and highlight how faults developed in weaker domains influence those developing in adjacent, relatively stronger material. We compare our 3D observations and analyses to previous analog and numerical modelling studies, relate our findings to the Great South Basin and other rift systems globally, and apply our observations to general continental rifting

concepts.

## 2 Numerical Approach

### 2.1 Modelling design and geometry

We model the 3D thermo-mechanical evolution of extending continental lithosphere using the mantle convection and lithospheric dynamics ASPECT (Kronbichler et al. 2012; Heister et al. 2017; Glerum et al., 2018; Naliboff et al., 2020; Pan et

al., 2022). The simulations span 500x500x100 km and fixed outward velocities drive extension at a constant rate of 5 mm/yr (Figure 2a). Inflow along the lower boundary balances outflow, while a stress free upper boundary allows the development of topography (Rose et al., 2017). Diffusion of the free surface at each time step minimises solver instabilities arising from localised deformation along faults, and acts as a coarse approximation of landscape evolution.

The initial lithospheric structure contains distinct lithologies with thermodynamic and rheological properties characteristic of

the upper crust (0-20 km depth), lower crust (20-40 km depth), and mantle lithosphere (40-100 km depth) (Figure 2a). The rheological structure follows a visco-plastic constitutive relationship, which captures both brittle (plastic) and ductile (viscous) deformation processes observed within rifts and rifted margins. Coupling brittle strain softening of cohesion and the internal





angle of friction with randomised initial plastic strain (IPS) enables the formation of distributed normal fault networks (Naliboff et al., 2017; Naliboff et al., 2020; Duclaux et al., 2020; Pan et al., 2022). We use variable distributions of the IPS along the

model length to define upper crustal volumes of differing strength (e.g., distinct geologic terranes), with the cohesion and angle of internal friction decreasing linearly between defined IPS values (e.g., strain softening interval).

The initial resolution throughout the model is set to 5 km, and refined to 1.25 km in the upper 20 km (i.e. upper crust) across the central 150 km of the model. This approach enables a relatively high resolution in the region of interest (upper crust), while producing 'natural' boundary conditions at its base. The full details of the model design and numerical methods are provided

in Appendix A, including the underlying governing equations.

## 2.2 Exploring upper crustal strength

We assign IPS values to 5 km³ blocks, termed Unit Blocks, in the Upper Crust across the central 150 km of the model, termed the Damage zone. IPS values were randomly assigned to unit blocks in a binary fashion, such that a block either has the minimum or maximum value specific to that strength. We define four 125 km wide upper crustal domains of varying strength,

oriented parallel to the extension direction (Figure 2a). From top to bottom, the domains are assigned Reference (Upper), Weak, Strong, and Reference (Lower) strengths (Figure 2a).

We generated four models with varying values and combinations of IPS in each of the domains. The initial cohesion (20 MPa) and internal angle of friction (30º), decrease by a factor of 4 between plastic strain values of 0.5-1.5. For each of our models we assign specified IPS values to Unit Blocks within the Damage Zone, with zero IPS outside of the zone. The assigned IPS

values vary between unit blocks in the Strong, Weak or Reference domains (Figure 2). The greater the value of IPS, and the greater the contrast in IPS between adjacent Unit Blocks, the weaker the domain. Unit Blocks in the Weak domain have values of 0.5 or 1.5 across all models. In Models 1 and 2, we characterise the strong domain as having zero IPS and the Reference domains as having IPS values of 0.5 or 0.75 (Model 1) and 0.5 or 1.0 (Model 2). In Model 3, the Strong domain is characterised by a constant IPS of 0.5 across all unit blocks within the domain, whilst IPS for unit blocks in the Reference domains are either

0.5 or 0.75. In Model 4, the Strong Domain is characterised by IPS of either 0.5 or 0.6 between different unit blocks, and a reference domain that varies between 0.5 and 1.0. In addition, we performed two runs of Model 4 using different randomised distributions of IPS between unit blocks, highlighting that whilst the randomised IPS may influence individual fault geometries, they do not affect the first order patterns identified in each domain.

## 3 Exploring strength parameter space

Each of our models is subject to the same 5 mm/yr extension rate. How strain localises across each model varies markedly between models and each of the domains. For each model, we now describe the overall fault geometries across the Weak, Strong, and upper and lower Reference domains (Figure 2a).





### 3.1 Model 1

Model 1 consists of a Strong domain with no initial strain perturbations and a relatively strong Normal domain. Strain rapidly
localises into the Weak domain, producing a well-defined fault network from early in the model run (Figure 3a). These high-
strain zones represent faults and interact with one another laterally, forming linkages and abandoned splays (Figure 3a). Outside
of this developed high-strain fault network the background strain is relatively low, forming strain shadows between the highly
localised faults. As the initial strain parameters for the Weak domain remain the same for each model, this domain is not
discussed for the other models, but will be examined in greater detail later.

The Strong Domain in Model 1 is characterised by zero strain weakening. At 10 My strain is distributed uniformly across the
model, with little localisation occurring. Some higher strain is observed at the edge of the strong domain in this model,
potentially representing edge effects at the edge of the Damage Zone as the entire domain forms a rigid, strong body that does
not extend.

The Reference domains in Model 1 are relatively strong compared to the reference domain in the other models. After 10 My
some localisation appears to have occurred in both Reference domains, with increased localisation occurring in the Upper
Reference domain (adjacent to the Weak domain) compared to the lower Reference domain, located adjacent to the Strong
domain. In the upper Reference domain increased localisation occurs close to the boundary with the weak domain. No clear
differentiation can be identified from top to bottom within the lower Reference domain.

### 3.2 Model 2

As with Model 1, no strain weakening occurs in the Strong domain of Model 2. Accordingly this produces similar patterns of
deformation to Model 1, with little localisation across the entire domain, and some increased strain observed at the boundaries
of the damage zone.

The Reference domain in this model is weaker than that of Model 1, with unit blocks assigned IPS values of 0.5 or 1.0.
Accordingly, the Reference domain in Model 2 shows increased localisation compared to Model 1 (Figure 3b). A similar
differentiation can be identified between the upper and lower Reference domains, with the former showing increased strain
localisation and better developed faults. Localisation does begin to occur in the lower Reference domain, with some increased
localisation seemingly occurring at the base of the domain (and the model) compared to the boundary with the strong domain
at the top of the domain (Figure 3b).

### 3.3 Model 3

In Model 3, the normal domain is characterised by the same IPS range as that in Model 1 (0.5-0.75), producing similar strain
patterns. In contrast to Models 1 and 2, the Strong domain in Model 3 is assigned a constant IPS of 0.5, with no variation
between adjacent unit blocks. Although strain weakening is permitted in the strong domain of this model, we still do not
identify any strain localisation. As with Models 1 and 2, where no strain weakening was present, the strong domain undergoes





relatively uniform strain, with some increased localisation at the edge of the domain where there is a contrast in IPS between
the outside of the model, where no strain weakening is present, and the damage zone where IPS is prescribed (Figure 3c).

## 3.4 Model 4

Model 4 is characterised by varying IPS values in the strong, normal and weak domains (Figure 2b). IPS varies between 0.5
or 1.5 in the weak domain, 0.5-1.0 in the Normal domain (as in Model 2), and, in contrast to the previous models, varying IPS
of 0.5-0.6 in the Strong domain. We performed two runs of this model, keeping the same values but changing how they were
distributed across the unit blocks prior to extension.

As in Model 2, strain localisation varies between the upper and lower Reference domains, with increased localisation occurring
in the upper Reference domain, adjacent to the Weak domain (Figure 3d). At the end of the model run, we begin to see some
strain localisation within the Strong domain with broad zones of increased strain beginning to develop (Figure 3d). We now
analyse this model in detail, examining the localisation of strain through time and in three dimensions.

## 165 4 Strain accommodation through time (Model 4)

Model 4 shows the greatest variation within each domain across the 10 My model run. We qualitatively and quantitatively
analyse the results of this model in three-dimensions at 2.5 My intervals (Figure 4).

### 4.1 Qualitative observations

The Weak domain is characterised by a network of widely-spaced, high-strain localised zones, separated by low-strain shadows
(Figure 4). Faults are sub-perpendicular to the extension direction, although some variation occurs, particularly at fault tips
(Figure 4a). In cross-section these faults form conjugate sets that join at the top of the ductile lower crust (Figure 4b). Fault
location and geometry, including the interactions and linkages between adjacent faults are established during the initial
timestep (100 Ky) of the model. Strain continues to be accommodated along this fixed network throughout the model run, as
evidenced by the fixed areas of high strain rate and non-initial plastic strain (Figure 4a; See Appendix B for gifs of the entire
model run).

Throughout the model run, the Strong domain is characterised by uniform strain with little to no localisation onto faults. Strain
rate across the model is highly variable, with high bands continually migrating across the area throughout the run and showing
no localisation (Figure 4a). Broad zones of elevated non-initial plastic strain strain start to develop towards the end of the run,
extending outwards from the boundaries of the Strong domain with the adjacent Weak and Reference domains (Figure 4b).
Both Reference domains display some evidence of strain localisation, intermediate between the complete localisation in the
Weak domain and the lack thereof in the Strong domain. Faults are typically sub-linear in plan view with limited interaction
occurring between adjacent structures. Looking at the strain rate, high strain rate structures display some transient properties
throughout the model run as faults move and strain does not fully localise onto established structures (Figure 4). Strain



localisation occurs diachronously between the lower and upper reference domains, with different degrees of localisation
occurring in each domain (Figure 4a). Increased strain localisation occurs in the upper Reference domain, with localisation in
the latter also occurring earlier in the model run. The upper Reference domain is adjacent to the highly localised Weak domain
whereas the lower Reference domain is adjacent to the Strong domain, which experiences little strain localisation.

In map-view, strain appears greatest in the centre of the faults in the Reference and Weak domains, decreasing towards the
fault tips (Figure 4a). High strain values are also identified along faults in the Weak domain close to its boundaries. The faults
are typically retarded at the domain boundary and strain rapidly decreases to background levels. In some instances, particularly
along the boundary between the Weak and upper Reference domain, faults may extend across the boundary, with lower
amounts of strain accommodated in the upper Reference compared to the Weak domain. In cross-section, faults display a high-
strain, highly localised core, decreasing to background levels away from the fault (Figure 4b). Faults appear more well-defined
in the centre of the Weak domain compared to the boundaries and other domains.

**4.2 Quantitative strain analyses**

To quantitatively analyse our model results we measured strain along a series of transects through Model 4. Cumulative strain
was measured across the centre of each of the domains and also at 2.5 km intervals covering the boundary between the Strong
and Weak Domains (Figure 5). Large vertical jumps in the cumulative strain show the location of faults, with the gap between
these jumps representing the fault spacing. A line displaying a constant gradient represents uniform strain across the model.
Individual faults may be difficult to distinguish, particularly those less localised structures. As such, measurements of
individual per-fault strain and fault spacing are approximate, although we can identify first-order differences between domains.
The total accumulated strain across the Strong Domain is less (~18) than that of the Weak Domain (~22) (Figure 5b). Although
both domains are subject to the same strain, relatively more strain is focussed into the Damage Zone in the weak domain than
in the strong domain. Throughout the model run, some strain is accommodated outside of the damage zone; strain is more
distributed between the 125 km wide damage zone and the rest of the model (375 km) in the strong domain compared to the
weak domain.

Faults remain fixed in position in the Weak Domain throughout the model run. These faults are spaced at 5-10 km intervals
with each fault accommodating strain of ~0.5-1. The areas between the faults are characterised by low-to-zero strain, of a
magnitude similar to the model areas outside the damage zone. In the Reference domains, some strain transfer and interactions
occur between fault tips, as identified using the strain rate parameter. Isolated faults appear to transiently link with adjacent
structures at various points throughout the model run, forming larger continuous structures (Figure 4c).

Faults within the Reference domain are more closely-spaced and accommodate less strain than those in the Weak domain. In
the lower Reference domain, per-fault strain is ~0.25-0.5 and spacing around 25 km. Values in the upper Reference domain
are relatively similar, with per-fault strains of 0.5-0.75 and spacing of 10-15 km. There are often less-developed faults between
the more localised structures in the Reference domains. We note a slight change in fault density across the upper Reference
domain, with more well-developed and localised structures, producing larger steps in the transects, present to the left side of





the model (Figure 5). This may in part reflect the distribution of the initial plastic strain within the upper Reference domain, or the prevalence of faults extending across the boundary from the adjacent Weak domain.

A key point is that fault spacing and per-fault strain, and therefore strain localisation, is highest in the Weak domain, decreasing into the upper Reference domain and the lower Reference domain. No localisation is identified in the Strong Domain, with no clear steps identified in the transect (Figure 5), which appears to approximate uniform strain (Figure 5b).

### 4.3 Strain accommodation across domain boundaries

We analyse fault geometry across the Reference-Weak and Weak-Strong domain boundaries (Figure 4). Strain is more localised in the Weak compared to the Reference domain, with the latter having a higher background strain (i.e. more distributed) (Figure 4a). The Weak-Reference boundary is characterised by a ~10 km zone of diffuse strain (Figure 4a). To the south, highly localised faults in the Weak domain are inhibited at and dissipate towards the Strong domain boundary (Figure 4b). Broad zones of elevated strain characterise the boundary-proximal Strong domain and persist up to 25 km away from the boundary (Figure 5c).

This transition from localised to diffuse strain across domain boundaries demonstrates a 'seeding' effect of faults in stronger domains by those in weaker domains (Figure 4). Established faults propagate into adjacent stronger domains, initially as broad zones of elevated strain that become increasingly localised. This may account for the lateral variation in fault density across the upper Reference domain; this reflects the density of faults in the Weak domain, with more seeding occurring to the left of the model (Figure 5a). Faults in the upper Reference domain are partially 'seeded' by those in the adjacent Weak domain. However, as the lower Reference domain is weaker than the adjacent Strong domain, faults here are not seeded and initiate independently, accounting for the differences between the Reference domains (Figure 3). In turn, faults in the lower Reference domain may, along with faults in the Weak domain, seed faults in the Strong Domain (Figure 4b).

### 4.4 Fault geometry

The fault network in the weak domain is established in the first timestep (100 ky) of the model. Faults within the network display a range of orientations and overall strain (Figure 5a, 6). Numerous relay ramps, lateral fault linkages and abandoned fault splays are present across the domain, with the larger faults associated with numerous synthetic and antithetic splays (Figure 6b). Further faults are also present at lower strain values, cross-cutting and oriented at a relatively high angle to the main fault (Figure 6a,b). These high-angle structures appear to roughly follow the distribution of initial plastic strain within the model (Figure 6b) and are cross-cut by the main fault in map- and cross-section view (Figure 6c). In some instances it seems that the cross-cut structures may also form antithetic/synthetic splays to the main structure (Figure 6b). This suggests the higher angle faults formed initially before being cross-cut by the main structure. This complex multi-generational faulting occurred within the first 100 ky of the model run (Figure 6).

At domain boundaries, faults are typically inhibited or greatly reduce in per-fault strain (Figure 6a). At the boundary between the upper reference and weak domains the main fault extends into the reference domain, albeit with lower overall strain (Figure



6a,c). In contrast, the main fault terminates at the strong-weak domain boundary. Strain is much more distributed within the
strong domain, although a wide zone of elevated strain associated with the fault can be identified (Figure 6a, c).

## 5 Discussion

### 5.1 Comparisons to previous numerical and analog modelling studies

In this study we present a series of 3D thermo-mechanical numerical models examining the influence of crustal strength on
rift development. Previous analog modelling studies highlight the influence and interplay between discrete structures
throughout the upper crust and lithosphere. Zwaan et al., (2021) show how arrays of intersecting discrete structures may
produce complex fault geometries. Similarly, Samsu et al., (2021) use analog models to show how discrete basement structures
can influence the location of later formed faults without reactivation. In our models we do not prescribe discrete individual
heterogeneities, instead modelling a diffuse zone of randomised weakness, thus our heterogeneities display no overall preferred
orientation (Figure 2). This is more representative of a pervasive weak rheology, similar to the analog models presented in
Samsu et al., (2021). A benefit of our numerical modelling approach is that we can assign weakness without relying upon
discrete heterogeneities or a weak seed. By altering the contrast between adjacent unit blocks we are able to effectively assign
varying bulk strengths to different domains.

Beniest et al., (2018) use analog modelling to investigate how lateral strength variations in the lithosphere influence rift style,
showing that strain preferentially localises into weaker areas with strong areas being resistant to strain and acting as rigid
blocks. Whilst our model supports the findings of Beniest et al., (2018) a key difference is that it is oriented at 90 degrees, with
each domain subject to strain rather than localised into one area. As a result of this we see some limited localisation within the
strong domain, particularly where faults begin to propagate across the domain boundary from adjacent domains (Figure 6).
Where either no strain weakening (Models 1 and 2, Figure 4), or no variation in strain weakening (Model 3, Figure 4) was
incorporated into the Strong domain, it acts as a rigid block with strain localising around its margins (Figure 4), in agreement
with the observations of Beniest et al., (2018).

We are able to identify along-strike changes in the rift physiography and examine how the faults behave at boundaries between
domains (Figure 5, 6). Both models are highly applicable to different geological areas, depending on the initial crustal
configuration and orientation of rifting (e.g. Brune et al., 2017).

Previously, numerical modelling studies have often examined first-order controls on the geometry and development of rifts
and rifted margins (e.g. Duretz et al., 2016; Naliboff and Buiter, 2015), rather than the three-dimensional upper crustal-scale
observations presented here. Some studies have attempted to replicate the complexity present within the crust and lithosphere
(e.g. Duretz et al., 2016) although these are typically limited to two-dimensions.  One of the key tenets of our study is the
ability to analyse rift development in 3D and to examine how rift physiography changes along-strike atop varying upper-crustal
properties. Numerous studies have also focussed on how deeper, lithospheric-scale, heterogeneities influence tectonic
processes (e.g. Heron et al., 2019; Schiffer et al., 2020). Gouiza and Naliboff (2021) use 3D numerical modelling to investigate



how lithospheric strength and thickness affects continental rifting and breakup in the Labrador Sea, although this focuses on the first-order rift margin geometry, rather than rift-scale faults and features identified in our models. Whilst we do not incorporate any heterogeneities in the mantle lithosphere in our models, the 150 km wide damage zone controls the first-order rift location, perhaps fulfilling a similar role to mantle heterogeneities during rifting (Figure 1c). Nevertheless, whilst deeper, lithospheric-scale structures may govern first-order rift location, we suggest that rift structural style and physiography is primarily controlled by upper crustal properties and structures.

Recently, a series of high-resolution 3D numerical investigations have explored fault network evolution and crustal-scale rifting processes (Duclaux et al., 2020; Naliboff et al., 2020; Jourdan et al., 2021; Pan et al., 2022). The models of Duclaux et al. (2020) and Jourdan et al. (2021) demonstrate the dynamics of normal fault network evolution during oblique rifting. Our models of orthogonal extension here share a similar setup to those of Pan et al. (2022) and build upon those of Naliboff et al., (2020), using a randomised distribution of initial plastic strain across a broad area rather than a single weak seed. These studies highlight how complex fault geometries may arise through fault interactions rather than individual heterogeneities. In contrast to these previous studies, our models highlight how fault geometries and rift physiographiers vary across crust of varying strength. The cross-cutting fault networks developed during the initial 100ky timestep of our models resemble transient fault networks generated in previous modelling studies that focus on fault evolution across smaller timescales (e.g. Cowie et al., 2000; Pan et al., 2022). We suggest that during the initial stages of extension faults exploit the weaknesses between the unit blocks, forming a complex fault network displaying a range of orientations. As extension progresses and strain increases during this initial 100ky timestep strain begins to localise onto larger faults oriented perpendicular to the regional stress (Figure 6). Similar observations have been made in nature; fractures that initially exploit non-optimally oriented fabrics at low strains may join to form larger through-going structures, as identified on shear zones in East Africa (Daly et al., 1989). In addition, the Ekitale Basin along the East African rift initially exploits low basement structures during low-strain extension, before being overprinted by more optimally-oriented structures (Ragon et al., 2019).

## 5.2 Comparison to natural systems and implications for rifting processes

Our models showcase an idealised scenario where rifting occurs parallel to crustal terranes of varying strength separated by vertical boundaries. Here, we relate key first-order observations from our models to other rift systems globally, including the Great South Basin, New Zealand. In addition, we draw upon our model observations to inform our understanding of rifting processes generally.

Rift structural style varies markedly between different domains; the Weak domain is characterised by a highly-localised and widely-spaced fault network that is established from the initial timestep in the model, whereas the Strong domain is characterised by a lack of localisation and relatively distributed strain (Figure 7). Beneath the Great South Basin, the Murihiku basement terrane is a dominantly sedimentary terrane sourced from a fore-arc setting (Tulloch et al., 2019; Sahoo et al., 2020) and is taken as a analog for the Weak domain in our models. The sedimentary nature of this terrane may cause it to be relatively weak and it also may contain a multitude of pre-existing weaknesses such as bedding planes and pre-existing faults (Tulloch




et al., 2019). Similarly, the Brook Street terrane is primarily composed of volcaniclastic material. In contrast, the Median

Batholith is predominantly granitic and relatively homogeneous, this is taken to represent an analog for the Strong domain in our models. Weaknesses are also likely present in the 'strong' Median Batholith terrane, for example, internal shear zones or the boundaries between individual granitic plutons (Allibone and Tulloch, 2004; Phillips and McCaffrey, 2019),. However, we suggest these heterogeneities are less pervasive and may display a lower difference in relative strength than heterogeneities in weak areas such as the Murihiku Terrane. Strong bodies, such as granites, typically resist strain localisation, as exemplified

by their role as 'blocks' in the 'block and basin' geometry of UK Carboniferous rift systems (Fraser and Gawthorpe, 1990; Howell et al., 2020), and the Sierra Nevada Batholith in the USA, which buffers extension in the Basin and Range (Ryan et al., 2020).

Strain localisation occurs diachronously across our model, first in the Weak domain, and last in the Strong domain (Figure 5a). This agrees with our assumptions of terrane strength from the Great South Basin, where rifting has been shown to initiate in

the 'weak' sedimentary/volcaniclastic Murihiku and Brook Street Terranes prior to the 'strong' granitic Median Batholith (Figure 1a) (Sahoo et al., 2020). Similarly, extension in the Tanganyika rift of the East African Rift system rapidly localises onto border faults where the rift traverses Proterozoic mobile belts, but remains distributed across the cratonic Bangwelu Block (Figure 1b) (Wright et al., 2020). Here, the mobile belts host prominent fabrics, forming weaknesses and acting similar to the large IPS contrasts in the Weak domain (Figure 2a), and thereby causing strain to preferentially localise. In contrast, the

cratonic Bangwelu Block hosts only weakly-developed fabrics, analogous to the small IPS contrasts present between Unit blocks in the Strong domain, inhibiting strain localisation (Wright et al., 2020). Similar observations have been made from analog modelling studies, which show more distributed (uniform) deformation in areas of stronger basement (Samsu et al., 2021).

Gouiza and Naliboff (2021), show how continental rifting and breakup in the Labrador Sea first occurred in the strong North

Atlantic Craton, before proceeding to relatively weaker adjacent basement terranes. On first consideration, this would appear to contradict our model results and the geological observations described above, with continental rifting and breakup proceeding rapidly in the relatively stronger areas of crust and suppressed in weaker areas (Gouiza and Naliboff, 2021; Peace et al., 2017). However, numerous heterogeneities are identified onshore in the 'strong' North Atlantic Craton and extend beneath the Labrador Sea (Peace et al., 2017; Wilson et al., 2006). We suggest that the presence of these sparse weaknesses

partition the 'strong', homogeneous body into multiple isolated 'islands of strength' separated by numerous weaker heterogeneities (Figure 7). Whilst the strong 'islands' resist extension, as in our model Strong domain (Figure 2), strain may rapidly localise along the surrounding weaker heterogeneities. Due to large differences in relative strength between the strong areas and intervening weaknesses, these areas may rapidly localise strain. At the rift-scale, strong bodies such as the Median Batholith beneath the Great South Basin or the North Atlantic Craton beneath the Labrador Sea may resemble homogeneous

areas similar to the Strong domain in our models (Figure 7). However, examining these areas in more detail highlights internal weaknesses that may localise strain and allow rift systems to propagate through these strong areas. Based on this, we suggest that the strong domains in our model may be more representative of these islands of strength within the rift.



At terrane boundaries, we find that faults are inhibited at boundaries with adjacent, stronger domains, before potentially propagating through (Figure 4). No faults are present in the strong domain, with faults arrested at its boundaries. However, faults do traverse the boundary between the weak and reference domains, albeit displaying lower strain in the relatively stronger domain (Figure 6). In the Great South Basin, we observe that faults commonly rotate into alignment with the boundary or segment and terminate against the stronger areas (Figure 1a) (Phillips and McCaffrey, 2019; Sahoo et al., 2020). Our model observations show that as faults are initially arrested at the boundaries, diffuse areas of strain form in the stronger area, potentially analogous to damage zones in nature (Figure 3b). We suggest that as extension progresses strain may continue to build up at the terrane boundaries, accommodated by localised structures in the weaker domains and diffuse strain in the stronger. These broad areas of elevated strain weaken the stronger domain, once it is sufficiently weakened, faults from adjacent domains may propagate through the domain boundary and 'seed' faults in the stronger domains (Figure 3a). These faults are then able to propagate through the stronger area. With further extension, the seeded faults in the strong domain may lead to the development of the islands of strength and intervening weaknesses (Figure 7). As strain continues to further weaken the established weaknesses, leading to a greater relative strength difference between them and the low-strain strong island. This creates a positive feedback cycle where strain is preferentially focussed into the weaker area (Figure 7).

## 6 Conclusions

We document characteristic structural styles associated with strong and weak crust and examine how strain is manifest across boundaries between areas of different strength. We relate our findings to multiple rift systems globally, offering insights into their evolution and to fundamental continental rifting processes.

We demonstrate that well-developed fault networks develop in weaker areas containing numerous heterogeneities whilst localisation is inhibited in relatively homogeneous, stronger areas. We find that strain initially localises in these weaker areas before eventually propagating into and traversing stronger areas (Figure 7), similar to observations from rift systems globally. Within the weak domain, multiple generations of cross-cutting faults develop in the first 100ky timestep, in agreement with other studies examining fault evolution across shorter time intervals. We show how the first-developed faults initially form at non-optimal orientations and follow the weaknesses present within the initial model setup. With continued extension in this initial timestep, the fault system reorganises with new faults aligning perpendicular to the extension direction and cross-cutting the older structures (Figure 6).

We also highlight how strain localisation and fault development is inhibited within the strong domain of our models. We find that faults are initially inhibited at the boundaries between different domains in our model, as they are at terrane boundaries in nature. Our models offer a temporal perspective however, showing that broad areas of elevated strain develop in the stronger areas adjacent to fault tips before the barrier is eventually breached and the fault can continue to propagate. The presence of faults within relatively weaker domains is able to seed the development of those in adjacent, stronger areas.



We are also able to highlight key differences between our idealised model observations and observations from geological
examples. Whilst the strong domain in our models represents a large relatively homogeneous body with only little relative
strength differences, similar examples in nature contain some weaknesses at the basin-scale that localise strain and may lead
to rapid fault development. We suggest that these weaknesses traverse the otherwise strong bodies, creating a series of isolated
'islands of strength' which are resistant to extension and more resemble the strong domain in our models.

Our modelling highlights how upper crustal strength distributions influence rift geometry and physiography. We relate our
findings to other modelling studies and rift systems globally and highlight key implications for our understanding of continental
rifting processes, particularly during early stages of rifting across a geologically complex crustal substrate.

## 7. Appendices

### Appendix A

We use the open-source, mantle convection and lithospheric dynamics code ASPECT (Kronbichler et al., 2012; Heister et al.,
2017) to model 3D continental extension (Fig. 1a) following an approach modified from Naliboff et al. (2020) and Pan et al.
(2022). The Stokes equations follow the incompressible Boussinesq approximation,

$$\nabla \cdot u = 0 \quad (1)$$

$$-\nabla \cdot 2\,\mu\,\dot{\varepsilon}(u) + \nabla p = \rho g \quad (2)$$

Where $u$ is the velocity, $\mu$ is the viscosity, $\dot{\varepsilon}$ is the second invariant of the deviatoric strain rate tensor, $p$ is pressure, $\rho$ is
density, and $g$ is gravitational acceleration.

Temperature evolves through a combination of advection, heat conduction, and internal heating:

$$\rho C_p \left( \frac{\partial T}{\partial t} + u \cdot \nabla T \right) - \nabla \cdot \left( \kappa \rho C_p \right) \nabla T = \rho H \quad (3)$$

Where $C_p$ is the heat capacity, $T$ is temperature, $t$ is time, $\kappa$ is thermal diffusivity, $\alpha$ is the linear thermal expansion coefficient,
and $H$ is the rate of internal heating.

Density varies linearly as a function of the reference density ($\rho_0$), linear thermal expansion coefficient ($\alpha$), reference
temperature ($T_0$), and temperature:

$$\rho = \rho_0 \left( 1 - \alpha \left( T - T_0 \right) \right) \quad (4)$$

The model domain spans 500 by 500 km across the horizontal plane (X, Y) and 100 km in the depth (Z) direction (Fig. 1a).
The grids are coarsest (5 km) on the sides and base of the model domain and are successively reduced using adaptive-mesh
refinement, increasing the resolution to 1.25 km over a central region measuring 190 x 500 x 20 km (Fig. 1a). Aside from
mesh deformation related to free surface evolution, the numerical resolution otherwise stays constant through time with no
further adaptive refinement steps. Broadly, this approach provides 'natural' boundary conditions for the formation of a





distributed fault network within the upper crust. Deformation is driven by prescribed velocities on the models side of 5 mm

       yr$^{-1}$, which are balanced by inflow at the base of the model.

       The model domain contains three distinct compositional layers, representing the upper crust (0-20 km depth), lower crust (20-

       40 km depth), and lithospheric mantle (40-100 km depth). Distinct reference densities (2800, 2900, 3300 kg m$^{-3}$) and viscous

415    flow laws for dislocation creep (wet quartzite; Gleason and Tullis, 1995, wet anorthite; Rybacki et al., 2006, dry olivine; Hirth

       and Kohlstedt, 2003) distinguish these three layers, which deform through a combination of nonlinear viscous flow and brittle

       (plastic) deformation (e.g., Glerum et al., 2018; Naliboff et al., 2020; Table A1).

       The initial temperature distribution follows a characteristic conductive geotherm for the continental lithosphere (Chapman,

420    1986). We solve for the conductive profile by first assuming a thermal conductivity of 2.5 W m$^{-1}$ K$^{-1}$, a surface temperature of

       273 K, and a surface heat flow of 55 mW/m$^2$, and constant radiogenic heating in each compositional layer (Table 1) that we

       use to calculate the temperature with depth within each layer. The resulting temperatures at the base of the upper crust, lower

       crust, and mantle lithosphere are 633, 893, and 1613 $^{\circ}$K, respectively. The temperature remains fixed at the top and base of the

       model, while the sides are insulating. The values of the compositional fields are only fixed at the base of the model.


       Rheological behaviour combines nonlinear viscous flow with brittle failure (see Glerum et al., 2018). Viscous flow follows

       dislocation creep, formulated as:

$$\sigma'_{II} = A^{-\frac{1}{n}} \dot{\varepsilon}_{II}^{\frac{1}{n}} e^{\frac{Q+PV}{nRT}} \quad (5)$$

       Above, $\sigma'_{II}$ is the second invariant of the deviatoric stress, $A$ is the viscous prefactor, $n$ is the stress exponent, $\dot{\varepsilon}_{II}$ is the second

invariant of the deviatoric strain rate (effective strain rate), $Q$ is the activation energy, $P$ is pressure, $V$ is the activation volume,

       $T$ is temperature, and $R$ is the gas constant. The viscosity (η) within ASPECT is calculated directly through

$$\eta = \frac{1}{2} A^{-\frac{1}{n}} \dot{\varepsilon}_{II}^{\frac{1-n}{n}} e^{\frac{Q+PV}{nRT}} \quad (6)$$

       with the resulting viscous stress equal to $2 * \eta * \dot{\varepsilon}_{II}$.

       Brittle plastic deformation follows a Drucker Prager yield criterion, which accounts for softening of the angle of internal

friction ($\phi$) and cohesion ($C$) as a function of accumulated plastic strain:

$$\sigma'_{II} = \frac{6C\cos\phi + 2P\sin\phi}{\sqrt{(3+\sin\phi)}} \quad (7)$$

       When the viscous stress exceeds the plastic yield stress, the viscosity is reduced such that the stress lies exactly on the yield

       plane (i.e., viscosity rescaling method; see Moresi and Solomatov 1998, Glerum et al. 2018).

       The initial friction angle and cohesion are 30° and 20 MPa respectively, and linearly weaken by a factor of 4 as a function of

finite plastic strain, which is derived from the second invariant of strain rate in regions undergoing deformation. The initial



plastic strain is partitioned into 5.0 km³ blocks that are randomly assigned binary values (ex: 0.5-1.5) in the different tectonic domains (see main text). This pervasive brittle damage field produces rapid localisation of a well-defined normal fault network (e.g., Pan et al., 2022).

Nonlinearities from the Stokes equations are addressed by applying defect correction Picard iterations (Fraters et al., 2019) to a tolerance of 1e-4 with the maximum number of nonlinear iterations capped at 100. The maximum numerical time step is limited to 20.

**Table A1**

|  | **Upper crust** | **Lower crust** | **Mantle lithosphere** |
|---|---|---|---|
| **Reference density** | 2800 kg m$^{-3}$ | 2900 kg m$^{-3}$ | 3300 kg m$^{-3}$ |
| **Viscosity prefactor (A\*)** | 8.57 x 10$^{-28}$ Pa$^{-n}$ m$^{-p}$ s-1 | 7.13 x 10$^{-18}$ Pa$^{-n}$ m$^{-p}$ s-1 | 6.52 x 10$^{-16}$ Pa$^{-n}$ m$^{-p}$ s-1 |
| **n** | 4 | 3 | 3.5 |
| **Activation energy (Q)** | 223 kJ mol$^{-1}$ | 345 kJ mol$^{-1}$ | 530 kJ mol$^{-1}$ |
| **Activation volume (V)** | - | - | 18 x 10$^6$ m$^3$ mol$^{-1}$ |
| **Specific heat (C$_p$)** | 750 J kg$^{-1}$ k$^{-1}$ | 750 J kg$^{-1}$ k$^{-1}$ | 750 J kg$^{-1}$ k$^{-1}$ |
| **Thermal conductivity (K)** | 2.5 W m$^{-1}$ K$^{-1}$ | 2.5 W m$^{-1}$ K$^{-1}$ | 2.5 W m$^{-1}$ K$^{-1}$ |
| **Thermal expansivity (A)** | 2.5 x 10$^{-5}$ K$^{-1}$ | 2.5 x 10$^{-5}$ K$^{-1}$ | 2.5 x 10$^{-5}$ K$^{-1}$ |
| **Heat production (H)** | 1 x 10$^{-6}$ W m$^{-3}$ | 0.25 x 10$^{-6}$ W m$^{-3}$ | 0 |
| **Friction angle (φ)** | 30 ° | 30 ° | 30 ° |
| **Cohesion angle (C)** | 20 MPa | 20 MPa | 20 MPa |


**Appendix B**

B1 – Left – Gif of Model 4 run showing the non-initial plastic strain interval across 10 My. Right – Model 4 run across 10 My showing the strain rate invariant attribute. See figures 4a and c for snapshots throughout the model run.

B2 – Cross section at y=375 km, showing the non-initial plastic strain attribute through 10 My of the model run. This location
corresponds to the boundary between the Weak and Reference (Upper) domains. See Figure 4b for 10 My snapshot.

B3 - Cross section at y=375 km, showing the strain rate attribute through 10 My of the model run. This location corresponds to the boundary between the Weak and Reference (Upper) domains. See Figure 4c for 10 My snapshot.

B4 - Cross section at y=300 km, showing the non-initial plastic strain attribute through 10 My of the model run. This section crosses the centre of the Weak domain. See Figure 4b for 10 My snapshot.

B5 - Cross section at y=300 km, showing the strain rate attribute through 10 My of the model run. This section crosses the centre of the Weak domain. See Figure 4c for 10 My snapshot.

B6 - Cross section at y=240 km, showing the non-initial plastic strain attribute through 10 My of the model run. This section is located 10km into the strong domain, proximal to the boundary with the weak domain. See Figure 4b for 10 My snapshot.

B7 - Cross section at y=240 km, showing the strain rate attribute through 10 My of the model run. This section is located 10km into the strong domain, proximal to the boundary with the weak domain. See Figure 4c for 10 My snapshot.

B8 – Cross section traversing all domains at x=250 km, perpendicular to those shown in B2-7. Cross-section shows the non-initial plastic strain. See Figure 4a for 10 My snapshot.

B9 – Cross section traversing all domains at x=250 km, perpendicular to those shown in B2-7. Cross-section shows the strain rate attribute. See Figure 4c for 10 My snapshot.

## 8. Code availability

The numerical simulations were run with the open source mantle convection and lithospheric dynamics code ASPECT (https://github.com/geodynamics/aspect), version 2.4.0-pre (commit 9355aec07). A copy of the ASPECT version used in this study is located with the Zenodo repository associated with this publication (https://doi.org/10.5281/zenodo.7317598).

## 9. Data availability

The parameter files required to reproduce the numerical experiments are contained with the Zenodo repository associated with this publication (https://doi.org/10.5281/zenodo.7317598). In addition, this Zenodo repository also contains the numerical solution files and postprocessing scripts used to create the images for each figure in this publication.

## 10. Executable research compendium (ERC)

Not Applicable

## 11. Sample availability

Not Applicable



## 12. Video supplement

Gifs of the model runs used throughout this manuscript are available in the supplementary material. The descriptions of each of these models are available in Appendix B.

**13. Supplement link: the link to the supplement will be included by Copernicus, if applicable.**

Not Applicable

## 14. Team list

Not Applicable

## 15. Author contribution

TP devised the conceptual model to examine and TP, JN, KM, and SP designed the experiments and model setup. The models were run by JN and TP. JN and SP developed automated model analysis tools. TP prepared the manuscript along with input from KM, JN and co-authors. All co-authors contributed to the writing, revising and preparation of the manuscript.

## 16. Competing interests

The authors declare that they have no conflict of interest

**17. Disclaimer**

Not Applicable

## 18. Special issue statement

Not Applicable

## 19. Acknowledgements

- The computational time for these simulations was provided under XSEDE project EAR180001
- We thanks Ake Fangerang and Guillaume Duclaux for constructive reviews on earlier versions of this manuscript





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



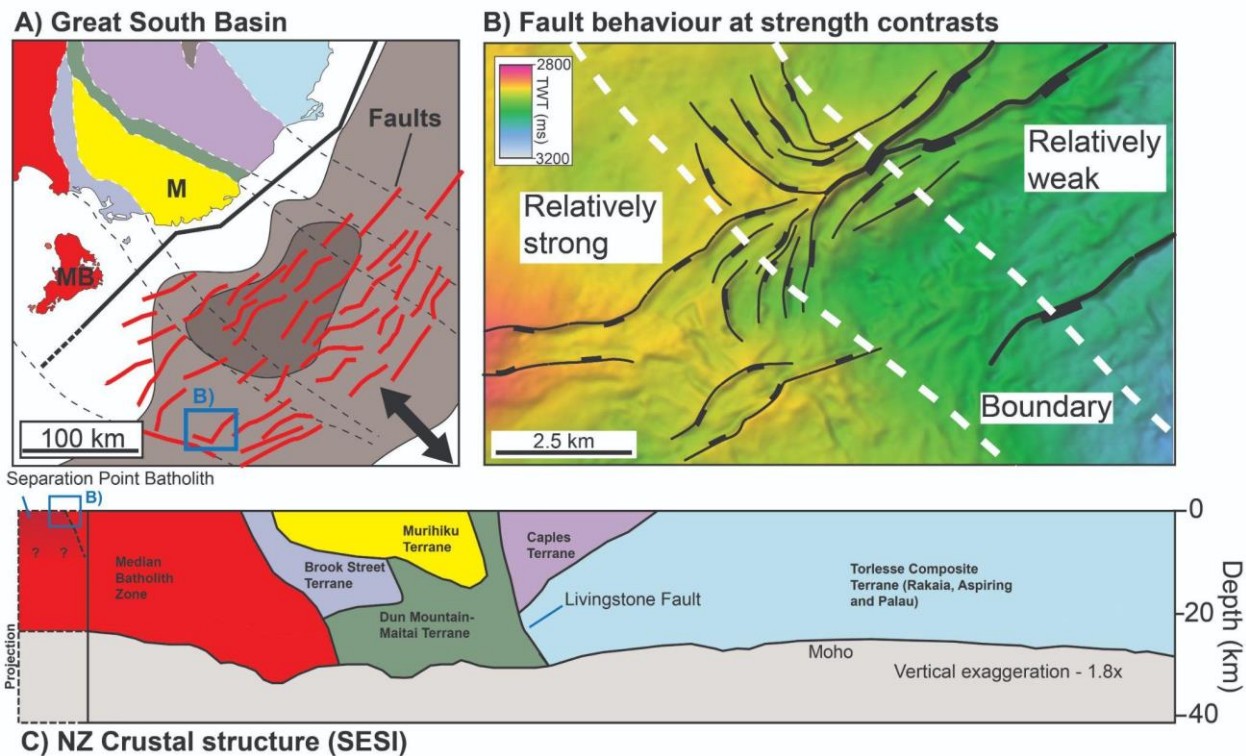

Figure 1: A) Map showing the basement terranes present beneath the South Island of New Zealand and their offshore projections beneath the Great South Basin. Offshore terrane projections after Ghisetti, 2010 and Mortimer et al., 2002. MB - Median Batholith; M - Murihiku Terrane. Main faults are shown after Sahoo et al., 2020. B) TWT structure map showing how faults splay and rotate along internal strength contrasts, after Phillips and McCaffrey, 2019. C) Generalised crustal structure offshore of the eastern coast of the South Island of New Zealand based on the SESI seismic survey, after Mortimer et al., (2002). The southern end of the line is projected to incorporate the Separation Point Batholith along the southern margin of the Median Batholith Zone, and the hypothesised location of panel B is also shown.



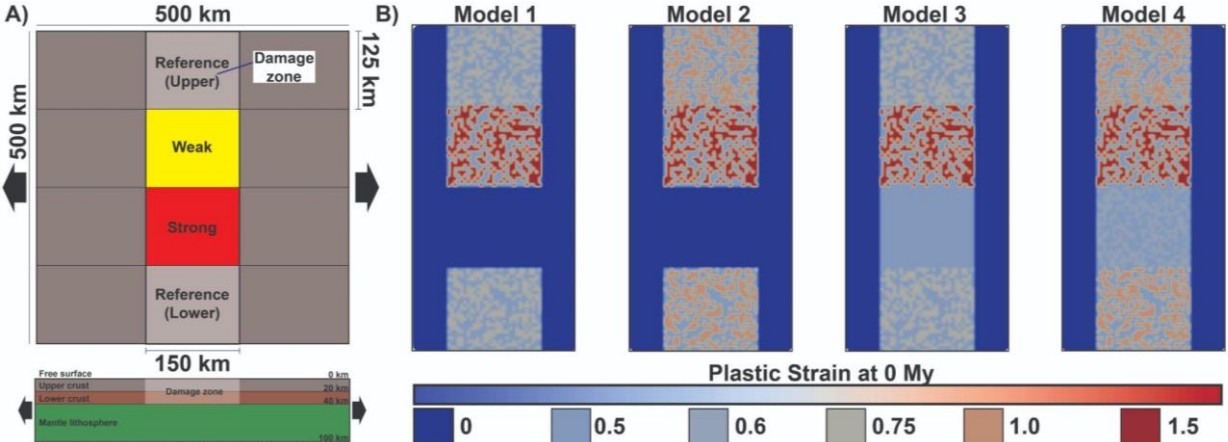

**Figure 2: A) Diagram showing the map view and cross-sectional setups for the model. Strain is applied across a 500 km wide and 100 km deep area, with a 150 km wide 'Damage Zone' extending to 40 km depth across the centre of the model where IPS is assigned to unit blocks. B) Images of the IPS distribution applied to each model prior to extension at 0 My. Variations in IPS values between 5 km3 Unit blocks define various strength terranes.**





Figure 3: Map view images of each model (A-D) showing the non-initial plastic strain (Left) and Strain Rate invariant (Right) attributes at 10 My.

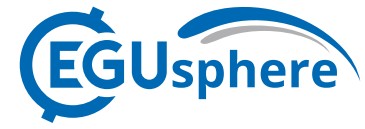

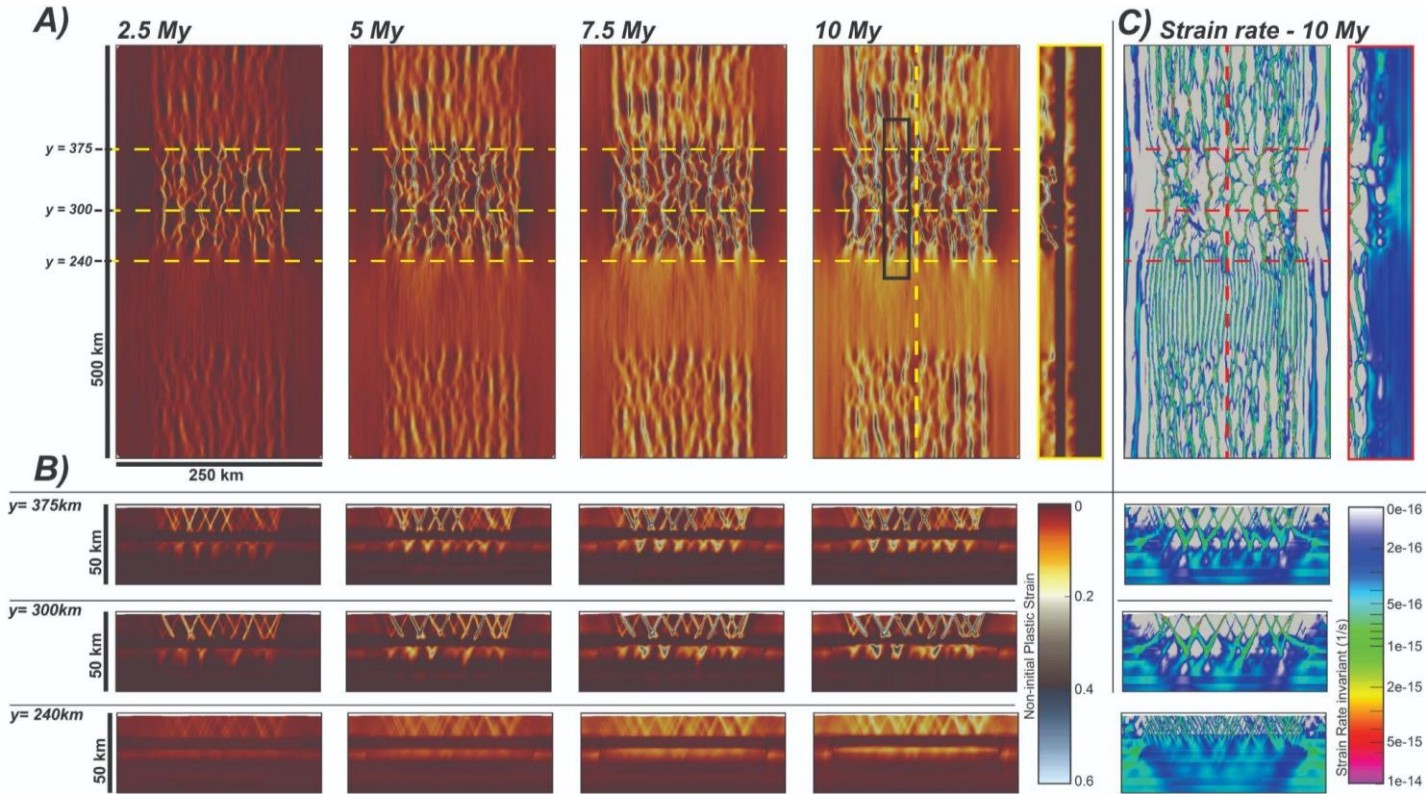

**Figure 4: A) Left - Map view images highlighting the fault evolution across the model using the Non-initial plastic strain attribute at 2.5 My, 5 My, 7.5 My and 10 My. The Black box shown on the 10My model timestep shows the location of Figure 6. B) Cross-sectional views across the centre of the Weak domain (y = 300 km); the boundary between the upper Reference and Weak domains (y = 375 km); and within the Strong domain 10 km from the Weak domain boundary (y = 240 km). C) Map and cross-sectional views of the strain rate attribute highlighting the fault network at 10 My.**



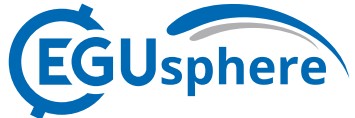

**Figure 5: A) Map-view image of non-initial plastic strain at 10 My across the central 250 km of the model. B) Strain analysis of multiple transects across the centre of each domain of the model, see A for location. The grey shaded area represents the location of the damage zone initially across the central 150 km of the model. Note how fault spacing and per-fault strain are greatest in the Weak domain whilst the Strong domain approximates uniform strain. C) Strain analysis of a series of model transects spanning the boundary between the Strong and Weak domains. Fault spacing and per-fault strain decrease across the boundary into the Strong domain.**





**Figure 6: A) Close-up map view image of a fault within the weak domain, see figure 4a for location. The locations of cross-sections and key faults are shown. B) Interpretation of fault geometries within and immediately adjacent to the weak domain. The interpretation and the distribution of initial plastic strain between unit blocks are shown with the interpretation. C) Cross-sections across the model, showing the cross-cutting fault systems. See A for location.**





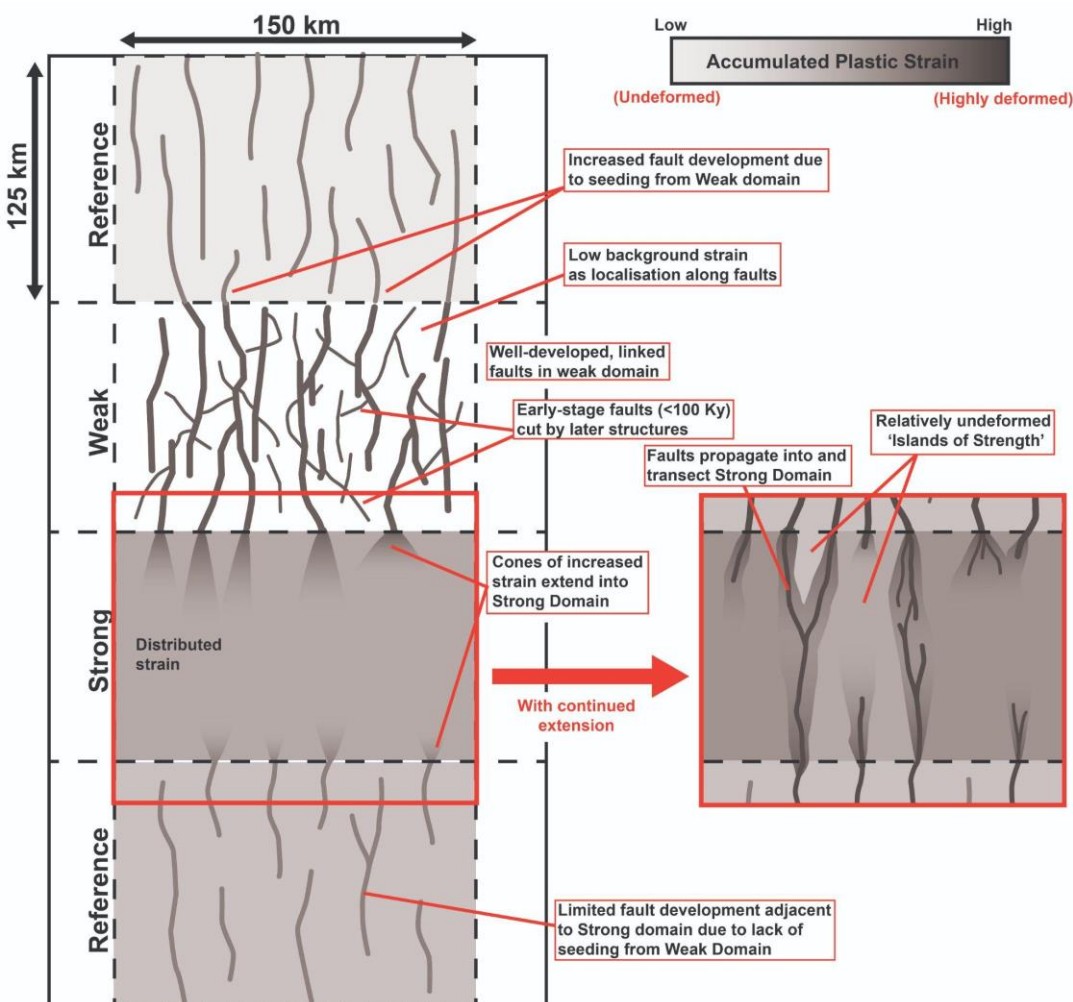

**Figure 7:** Cartoon showing the characteristic structural styles associated with each domain and the key concepts demonstrated in our model. Cones of increased strain extend into the strong domain. As extension progresses, faults would eventually propagate into the Strong domain forming throughgoing structures that localise strain and relatively undeformed islands of strength.