# Peer review of "The influence of crustal strength on rift geometry and development – Insights from 3D numerical modelling"

_EGUsphere, 2022_

## Author Comment (AC1)

**Reviewer 1 – Anonymous referee**

This manuscript addresses the very interesting question on the effect of along-strike strength variations on fault localisation processes. It uses state-of-the-art 3D numerical models that are complemented by observations from several examples of rifts worldwide but especially by geological observations of the Great South Basin, New Zealand, that features particularly prominent along-strike lithological variations. The results are well discussed and linked to previous insights. I find the interaction of strain localisation processes across terrane boundaries particularly interesting and well balanced. I only have a few minor suggestions that hopefully help to improve this interesting paper even more.

*We thank the reviewer for their positive comments and have modified the manuscript accordingly. Please find detailed responses to the individual comments below. Line numbers correspond to the track changes final document.*

*** Comments:

Fig. 1: An overview map could provide the reader with the plate tectonic context and the location of the study region relative to New Zealand. It would be useful to annotate New Zealand on the map in panel A. The cross section in panel C should be annotated in the map view of panel A. I suspect it is the black line, but I can't be certain.

*We have now added an inset to panel A showing the regional geological setting of New Zealand and the location of the study area. The location of the section shown in Panel C has also been annotated in A, corresponding to the black line.*

It would be good to motivate the model setup a bit more with the natural example. It's absolutely fine if some of the employed values are generic, but this should be mentioned. Here are some points that could be addressed:

- Line 85: is the 5 mm/yr extension velocity based on local divergence rates? If yes this would be good to mention here.

*The 5mm/yr extension rate used in our models are not based on the chosen example. The concept and geometry of our models are based on the study area, but the extension rate is based on an average of velocities used in other recent 3D models of continental rifts (Naliboff et al., 2020; Gouiza and Naliboff, 2021; Pan et al., 2022) and many previous 2D numerical models. Given insufficient data exists to carefully assess strain rates in the region, we decide it would be best to use an average (or median) velocity used in previous rifting studies and is also close in magnitude to active continental rifts that have experienced similar amounts of extension. A sentence describing this has been added to the text (Line 86-87).*

*The findings of Pan et al. (2022) suggest that increasing or decreasing the velocity by a factor of 2 is likely to affect second order features of the fault network (i.e., number of faults and displace-offset distributions) but we do not anticipate it would change the key findings of this study in regards to how faults interact with rheological boundaries. However, we anticipate it would affect the timing of when faults in our different model domains (normal, strong, weak) propagate into neighbouring domains. For example, increasing the extension rate would likely promote faster localization onto*

*fewer faults within the weak zone, accelerated formation of small faults within the strong domains, and subsequent propagation of major faults across the strong domain.*

- Line 90: Are layer thicknesses consistent with the chosen example?

*We base the map view geometry and the distribution of areas of differing strength on the Great South Basin. Crustal thicknesses in the model are based on general thicknesses of unstretched crust. We have clarified this in the revised manuscript (Line 92).*

- Line 101: Why do you choose 5 km^3 blocks?

*The choice of 5 km was based on Pan et al. (2022), which found using a block size of at least 2-4x the numerical resolution (1.25 km here) was required to sufficiently localise deformation at the onset of extension (Line 109-111). The findings of a follow up study (Pan et al., In Revision; Preprint https://doi.org/10.31223/X5G65Q) suggests that increasing or decreasing the block size by a factor of 2 is likely to affect fault spacing, but we do not anticipate this would have a first-order control on the results of this study.*

*** Minor details:

Abstract and text in general: double check capitalisation of "weak", "strong" and "initial plastic strain".

*We have now checked and modified the manuscript to ensure that Weak, Strong, Reference and Initial Plastic Strain are correctly capitalised throughout. Weak, Strong and Reference are capitalised when referring to domains.*

Line 16: "extension occurs perpendicular to distinct geological terranes and parallel to terrane boundaries" I find the description of extension "perpendicular to terranes" but "parallel to terrane boundaries" misleading. I suggest to delete "perpendicular to distinct geological terranes and".

*Have modified this to say that extension occurs parallel to the boundaries between distinct geological terranes (Line 18).*

Fig. 4: Cross sections are vertically exaggerated. It would be useful to note in the caption by how much.

*This has now been added to the figure captions where appropriate. The cross-sections in Figure 4 are shown at 2.5x vertical exaggeration, whereas those in Figure 6 show no V.E.*

Section 5.1: There is a recently submitted analog/numerical modelling manuscript at Solid Earth Discussion by Schmid et al. (https://doi.org/10.5194/egusphere-2022-1203) that might be relevant to this section.

*We agree that this is relevant to this section and have added a sentence on this in the revised manuscript (Line 278).*

Some figures contain rather small font that could be enlarged for better readability.

*Figures have been modified to ensure that the text is clear and legible.*

Citation: https://doi.org/10.5194/egusphere-2022-1278-RC1

---

## Author Comment (AC2)

**Reviewer 2 – Guillaume Duclaux**

Review of "The influence of crustal strength on rift geometry and development – Insights from 3D numerical modelling", by T. Phillips, J. Naliboff, K. McCaffrey, S. Pan, J. van Hunen, and M. Froemchen.

The manuscript by Phillips et al. presents new 3D thermomechanical models of continental rift development and propagation in heterogeneous crustal domains. Such models are new and take advantages of advanced numerical developments. The manuscript is well written, well organised and the objectives are clear, i.e. to mimic surface fault patterns in rifts developed across continental regions with varying mechanical properties. The variation of the 'strength' of the crust in the models is set by imposing various range of initial frictional-plastic damage in adjacent weak regions of the continental crust layer. This work nicely illustrates the process of faults propagation and connectivity during rifting in heterogeneous regions. It is of broad interest to the tectonic community, either to those studying fault network development, or for improving our understanding of rift systems and passive margin evolution in regions with strong tectonic inheritance (i.e. pretty much everywhere).

The paper is well written and organised. I reviewed an earlier version of this work (submitted to another journal in 2021) and I am pleased to write that this version is substantially improved. I believe the manuscript is of broad interest to Solid Earth readership. It needs mostly editorial polishing, and I recommend accepting this contribution with minor revisions.

I present below a list of comments and questions that I believe should be addressed to improve the quality of the contribution:

- line 85: Is the extension velocity full rate or half rate? Could you please precise if the 5 mm/yr is applied on each wall or is it 2.5 mm/yr on each wall?

*The extension rate of 5 mm/yr is a full extension rate, with 2.5 mm/yr applied on each side wall. This has been clarified in the text on lines 85-87*

- There are many naming or spelling inconsistencies for the domain names. It should be "Reference domain" in line 124, 150, 157. Please make sure the case for the domain names is consistent throughout the manuscript (Strong Domain vs. Strong domain vs. strong domain, weak domain, Weak domain, etc.). I have highlighted in yellow some of the mistakes I found in the attached pdf.

*This has since been corrected in the revised manuscript.*

- lines 135-138 (and elsewhere). A suggestion: northern and southern could be a better wording than upper and lower… upper / lower have a vertical connotation, like the upper and lower crust. Because you discuss results from 3D models, I believe it brings some confusion. Same goes with terms like "top" (e.g. line 148).

*We agree with the comments from the reviewer here and have modified the revised manuscript to refer to the North and South Reference domains rather than upper and lower.  Top and Bottom are also now only used to refer to vertical features. Compass directions are used to refer to describe map-view observations.*

- line 141: "increased strain", is it the total finite strain, or the cumulative frictional plastic strain, or the instantaneous strain (i.e. strain rate), or all of those?

*Here we are referring to the cumulative frictional plastic strain. This has been clarified in the text on lines 128-129*

- line 155: I'm curious here. Is the material rheology different between the central domains and the lateral regions? As in there is no frictional plastic behaviour. Or did you just not impose any pre-existing frictional plastic strain, and yet the material could still soften?

*In this case the rheology (flow laws, plastic strain weakening intervals) is identical between the domains and all the materials can still soften. The only difference between the domains is the degree of initial plastic strain the 5 km blocks in each domain contain, and thus their "starting point" along the plastic strain weakening interval. We have revised the text on this line (now line 163) to indicate that the only differences between these regions is the amount of initial prescribed plastic strain. We have also revised section 2.1 (lines 98-99) to reinforce that the only differences between distinct model domains is the amount of initial plastic strain.*

- sub-Figures calls are lower case in the text, but upper case in the figures and captions (2b vs. 2B, 3d vs. 3D).

*Have changed this in the figure captions.*

- line 169: I know that section is qualitative but could you provide a range for the "widely-spaced" network? Is the 10-15 km distance written line 214 the actual spacing value?

*Fault spacing is highly variable across the Weak domain, but the main, high-strain structures do form a more widely-spaced network. We have included a range of 10-15 km here, based on information from Figure 5.*

- line 177: "high bands continually migrating": Could you provide more information on those structures? What's that if not localisation… is it a transient localisation phenomenon? Or a feature of the model? I find it a little worrying.

*We have since changed this sentence in the revised manuscript (Line 186-189). The high-value bands that migrate across the model represent the instantaneous strain rate. As this is constantly migrating, no localisation occurs and strain does not accumulate in any area, hence no localisation.*

- line 192-194: Because of the strong vertical exaggeration of the sections in Fig 4B) I find it difficult to assess the dip angle of the structures. It would be a nice addition in the 3D description here.

*We agree with the reviewer here and have added a description of the fault dip of 45-50 degrees (Line 206). Following a comment from reviewer 1 we have also included the vertical exaggeration in the figure captions.*

- line 196: "measure strain", again is this the frictional plastic or total strain?

*This refers to the non-initial plastic strain. We now explain this more clearly in the revised manuscript (Line 210).*

- line 202: I find this method quite interesting to measure the frictional plastic strain. How do this number (18 and 22) compare to the horizontal stretching factor of the model at the same timestep?

*The horizontal stretching factor is the same across the model. This method identifies the amount of strain that is focussed into the central damage zone rather than the total extension that has occurred across the entire 500 km model width. We have modified this section in the revised manuscript to reflect this (Line 218-222).*

- line 207: "Faults remain fixed": That can't be right! As the model stretches the faults should also advect slightly at the surface.

*We agree with the point raised by the reviewer. The geometry and topology of the fault network remains relatively fixed in the weak domain throughout the run. We have since adapted this sentence to state: "The fault network maintains its overall pattern and topology in the Weak Domain throughout the model run, with individual faults displaying some advection outwards as the model extends." (Line 223-224).*

- You don't seem to discuss the hourglass shape of the fault network with a neck in the Weak domain. Any comment on this?

*We suspect this may be due to some combination of the manner in which faults intersect at depth, length of the weak zone, and fault linkage across the domains. For example, we did not observe this pronounced hourglass behaviour in Pan et al. (2022) where the rift zone was 180 km in length and truncated in either end at an effective "strong zone", although that study was conducted at higher resolutions (625 m). We prefer not to speculate this in the text as we would need to run a few additional tests to make an informed comment, but we do plan to investigate further at some point in the near future.*

- line 210: "transiently". May I suggest "progressively" instead? If it was transient after linking the faults would unlink… I doubt this is the case.

*Agreed – this has been changed in the revised manuscript.*

- line 221: Yet in the cross sections in figure 4 there seem to be some localisation in the northern part of the Strong domain. Which threshold do you use to decide whether strain is localised or not? In a section about quantitative analysis, I would expect to read a few more numbers.

*We agree with the reviewer that some localisation is beginning to occur in the northern part of the strong domain. These zones of increased strain represent the extensions of localised faults in the weak domain into the strong domain. We do not designate a specific threshold to determine whether*

*strain is localised or distributed in the model. In these simulations multiple criteria would likely be needed to quantitatively classify the degree of strain localization (strain rate, fault spacing, etc.), but this would likely require a separate paper building on the methods used in Pan et al. (2022, G3). The strain rate shows migrating high-value bands where strain is not localised. We have added a sentence on this to the text (Line 189, 240-241).*

- line 288, 289: Jourdon, spells with an "O" - it is spelled correctly in the references though.

*This has been corrected in the revised manuscript (Line 288)*

- line 309: the term "initial timestep" bothers me a bit… it is the first output, but many calculation timesteps have taken place, right? At this resolution and with such imposed lateral velocities and thermal properties I imagine that timesteps must be of the order of 5 to 10 kyr, isn't it?

*We agree with the reviewers comments, and have modified the text accordingly to indicate the fault network is well established by 100 Ky rather than the initial timestep (Line 315). The readers are referred to the supplementary animations to view this directly. The time steps in our models are 20 Kyr (Appendix A).*

- line 338-341: Or is it related to mantle rheology rather than the upper crust? I am not fully convinced yet of the statement that rift physiography and structural style are primarily controlled by upper crustal properties (lines 285-286) ...

*We agree with the reviewer that this point required clarification. We have modified the text to indicate that the style of rifting and breakup in the Labrador Sea (and many other rifts) is often controlled by the first-order lithospheric structure (i.e., crustal thickness, rheological layering, geotherm). The control of brittle crustal inheritance proposed is likely to be a second order effect, but absent any variations in lithospheric structure may act as a first-order control in determining variations in deformation style between domains (as observed in our focus area). Lines 356-365 have been modified to reflect these points.*

- line 366-368: There surely is an upper limit in how weaken a material can be? In your models this is the max frictional plastic value for the weakening function. After this value is reached could faults be abandoned and new one created? I suppose it will strongly depend how the fault dip evolves… as stretching accumulates the fault plane should rotate to become less steep, and thus less optimally oriented. Do you have any comment on this? Or maybe in the Myr of the models it isn't obvious that the dip angle evolves for the faults… can't be easily seen from the figures as the vertical exaggeration in the cross-section is very significant.

*We agree with the reviewer in that there is absolutely an upper limit to how much a material can weaken in the brittle domain. Realistically, the magnitude of brittle weakening we have imposed here (4x) is probably at the uppermost reasonable limits. Regarding fault abandonment, we have observed in previous studies (Naliboff et al., 2017; Peron-Pinvidic and Naliboff, 2020) that with sufficient extension well established faults are rotated to sufficiently low angles that they deactivate and are subsequently incised. However, that process was observed at the later stages of continental*

*breakup rather than the initial stages modelled here. We suspect that with continued extension this process would occur in our models, although we suspect it is somewhat controlled by the thickness and relative strength of a mid-crustal ductile layer*

- line 369-370: If this is one timestep, how can you have multiple generation of features associated with it? It comes back to my previous question about timestepping of the code vs. of the outputs.

*Indeed, the reviewer is correct and we should not have used the term "one timestep" here. Rather this simply represents x numerical timesteps (20 Kyr each) between when we output values. The text has been revised accordingly.*

- Table A1: What is p exponent in the viscosity prefactor? Please edit the baseline to superscript for -1 in s^{-1}.

*In this case the p exponent refers to the grain size exponent, which is 0 for the dislocation creep flow law*

- Figure 1: Would be nice to add a grid or some coordinates at least for A)

Adding some orientation SW - NE to figure C) would be good too.

*We have now added orientation to panel c) in the figure. Following a comment from Reviewer 1 we have now included an inset showing the geological setting of New Zealand. As the model is based on general characteristics of the area, we suggest that this aptly shows the regional location of the study area.*

- Figure 3 and 4: colour scales titles could be modified slightly: + Non-initial "cumulative" Plastic Strain; + Strain rate "second" invariant; + the min scale value should for $\varepsilon_{ii}$ should be < 1e-16 rather than 0e-16.

*These have been changed in the revised manuscript.*

- Figure 7: like in the text, the case of Strong/strong, weak/Weak, Domain/domain should be cleaned-up for consistency.

*This has been corrected in the revised manuscript*

Guillaume Duclaux,

Nice, 02/01/2023

Citation: https://doi.org/10.5194/egusphere-2022-1278-RC2